# The TRPV1 Receptor Is Up-Regulated by Sphingosine 1-Phosphate and Is Implicated in the Anandamide-Dependent Regulation of Mitochondrial Activity in C2C12 Myoblasts

**DOI:** 10.3390/ijms231911103

**Published:** 2022-09-21

**Authors:** Sara Standoli, Sara Pecchioli, Daniel Tortolani, Camilla Di Meo, Federico Fanti, Manuel Sergi, Marina Bacci, Isabelle Seidita, Caterina Bernacchioni, Chiara Donati, Paola Bruni, Mauro Maccarrone, Cinzia Rapino, Francesca Cencetti

**Affiliations:** 1Faculty of Bioscience and Technology for Food Agriculture and Environment, University of Teramo, 64100 Teramo, Italy; 2Department of Experimental and Clinical Biomedical Sciences Mario Serio, University of Florence, 50121 Firenze, Italy; 3European Centre for Brain Research (CERC)/Santa Lucia Foundation IRCCS, 00143 Rome, Italy; 4Department of Biotechnological and Applied Clinical Sciences, University of L’Aquila, 67100 L’Aquila, Italy; 5Faculty of Veterinary Medicine, University of Teramo, 64100 Teramo, Italy

**Keywords:** transient receptor potential vanilloid type 1, methanandamide, sphingosine 1-phosphate, mitochondrial membrane potential, C2C12 myoblasts

## Abstract

The sphingosine 1-phosphate (S1P) and endocannabinoid (ECS) systems comprehend bioactive lipids widely involved in the regulation of similar biological processes. Interactions between S1P and ECS have not been so far investigated in skeletal muscle, where both systems are active. Here, we used murine C2C12 myoblasts to investigate the effects of S1P on ECS elements by qRT-PCR, Western blotting and UHPLC-MS. In addition, the modulation of the mitochondrial membrane potential (ΔΨm), by JC-1 and Mitotracker Red CMX-Ros fluorescent dyes, as well as levels of protein controlling mitochondrial function, along with the oxygen consumption were assessed, by Western blotting and respirometry, respectively, after cell treatment with methanandamide (mAEA) and in the presence of S1P or antagonists to endocannabinoid-binding receptors. S1P induced a significant increase in TRPV1 expression both at mRNA and protein level, while it reduced the protein content of CB2. A dose-dependent effect of mAEA on ΔΨm, mediated by TRPV1, was evidenced; in particular, low doses were responsible for increased ΔΨm, whereas a high dose negatively modulated ΔΨm and cell survival. Moreover, mAEA-induced hyperpolarization was counteracted by S1P. These findings open new dimension to S1P and endocannabinoids cross-talk in skeletal muscle, identifying TRPV1 as a pivotal target.

## 1. Introduction

Sphingosine-1-phosphate (S1P) is a lysosphingolipid that derives from the hydrolysis of ceramide and the subsequent phosphorylation of sphingosine, catalyzed, respectively, by multiple ceramidases and sphingosine kinase 1 (SphK1) and 2 (SphK2) [1,2].

Once generated, S1P can be degraded by S1P lyase (SPL) [3,4], or alternatively recycled into ceramide through S1P phosphatases (SPP) and ceramide synthases. In addition, S1P can be released outside the cell via unspecific transporters, such as ATP-binding cassette transporters and the specific Spinster homolog 2 [5], and MFSD2B [6] carriers. All these metabolic pathways are necessary to maintain the sphingolipid biostat, crucial for a wide range of physiological processes, including determination of cell fate [7]. Once exported outside the cell, S1P can bind to five different G-protein coupled receptors, called S1P1-5 [8]. These S1P receptor subtypes are coupled to multiple G proteins and are differently expressed in the various cell types; thus, they can activate distinct signaling pathways, which, in turn, drive specific biological processes [9]. Consequently, S1P receptors impact on biological processes such as angiogenesis, immune response, tumorigenesis, embryonic development and skeletal muscle properties [8,10].

Of note, most of the biological processes under S1P control are also regulated by the endocannabinoid system (ECS) [11,12], which comprises the bioactive lipids family of endocannabinoids (eCBs) and a complex array of their receptor targets and metabolic enzymes necessary for their biosynthesis and degradation [13,14]. The first eCBs that were discovered, and that remain the most widely studied to date, are N-arachidonoyl ethanolamine (AEA) and 2-arachidonoyl glycerol (2-AG) [15].

AEA is the ethanolamide of arachidonic acid, produced mainly through the catalytic activity of N-acylphosphatidylethanolamine-specific phospholipase D (NAPE-PLD), and degraded principally by fatty acid amide hydrolase (FAAH) [16]. 2-AG is often the most abundant eCB in human cells, tissues and fluids, and is produced by diacylglycerol-lipases α and β (DAGL α and β), to be then hydrolyzed principally by monoacylglycerol lipase (MAGL) [13,17]. Both AEA and 2-AG bind primarily to G protein-coupled type 1 and 2 cannabinoid receptors (CB_1_ and CB_2_), which are expressed throughout the human body [18]. It is now apparent that eCBs can also bind to non-CB_1_/non-CB_2_ receptors, such as the orphan G-protein-coupled receptor GPR55 [19,20,21], and the transient receptor potential vanilloid 1 (TRPV1) channel [22]. Each receptor triggers a distinct signal transduction cascade, as detailed elsewhere [12,18,20,21,22,23].

Interestingly, S1P and eCB signaling pathways could interact with each other [24]. Not only S1P receptors and CB_1_/CB_2_ share 20% sequence identity [25], but also the activation of CB_1_ regulates sphingolipid metabolism with ceramide accumulation [26]. For instance, (i) both AEA and S1P act synergistically to regulate rat coronary artery reactivity [27], (ii) AEA treatment increases the phosphorylation of SphK1, and (iii) AEA-mediated vasorelaxation is attenuated by inhibition of SphK1 and SphK2; notably, the latter effect of AEA engages S1P3 [27]. More recently, it has been demonstrated that AEA can modulate SphK1 to increase the generation of S1P, thus mediating changes in blood pressure through S1P1 [24]. Of interest, the non-selective S1P receptor modulator FTY720 (2-amino-2-(2-[4-octylphenyl]ethyl)-1,3-propanediol), as well as endogenous sphingosine, have been shown to dose-dependently inhibit in vitro binding of selective CB_1_ antagonists and agonists to ectopically expressed mouse CB_1_ in CHO-K1 cells [28]. Yet, clear evidence of a possible cross-talk between S1P and eCB systems remains elusive. Here, we sought to fill this knowledge gap by interrogating the effect of S1P on eCB signaling in murine C2C12 myoblasts that express both systems [29,30,31]. In this context, it should be recalled that metabolism and signaling of S1P were found to be involved in skeletal-muscle biology [31,32,33,34], and that metabolism and signaling of eCBs have been shown to control skeletal muscle cell differentiation [35] along with myometrium contractility [36]. Thus, here we ascertained the possible modulation of the main ECS elements at gene and protein expression levels, along with the content of AEA and 2-AG in C2C12 myoblasts challenged with S1P. Moreover, due to the effect of AEA on energy homeostasis [37,38,39], mitochondrial membrane potential (ΔΨm) was measured in C2C12 cells treated with the non-hydrolyzable AEA analogue methanandamide (mAEA) [40], in the presence of S1P, as well as of agonists and antagonists to eCB-binding receptors.

We provide unprecedented evidence that S1P selectively increases TRPV1 expression in skeletal muscle C2C12 cells, and that TRPV1 transmits a dual action of mAEA depending on its concentration: at high dose (10 μM) mAEA decreases cell survival and inhibits mitochondrial activity, whereas at low doses (<5 µM) it increases mitochondrial membrane potential and induces the expression of peroxisome proliferator-activated receptor gamma coactivator 1-α (PGC1α), a transcriptional coactivator of genes implicated in the regulation of mitochondrial biogenesis and function [41]. Overall, this study discloses a new modulatory role of S1P in skeletal-muscle cells, which engages a distinct ECS element and supports the concept that different bioactive lipids can indeed cross-talk with each other.

## 2. Results

### 2.1. Effect of S1P on the Expression of ECS Elements

The mRNA expression of eCB-binding receptors (CB_1_, CB_2_, GPR55, TRPV1) and AEA and 2-AG metabolic enzymes (NAPE-PLD, FAAH, DAGLα, DAGLβ, and MAGL) was evaluated by qRT-PCR in C2C12 cells treated for 24 and 48 h with S1P at 1 μM, a dose that was shown to be effective to induce myogenic differentiation [32]. Upon exposure to S1P for 24 h, CB_1_ and CB_2_ mRNA expression showed a trend towards increase, yet not statistically significant, whereas GPR55 and TRPV1 mRNA levels were significantly enhanced (*p* < 0.01 vs. control) (Figure 1a). FAAH, DAGLα and DAGLβ were also significantly increased (*p* < 0.01 vs. control, for FAAH and DAGLβ; *p* < 0.05 vs. control, for DAGLα), unlike NAPE-PLD and MAGL that remained unaffected (Figure 1b).

Such an increase in selected elements of the ECS returned to control levels when S1P exposure was prolonged for additional 24 h (Figure 2a,b), with the exception of GPR55 gene expression, which was significantly reduced (*p* < 0.01 vs. control) (Figure 2a).

Then, the protein expression of eCB-binding receptors, AEA and 2-AG metabolic enzymes was evaluated by Western blot in C2C12 cells treated with 1 μM S1P for 24 h. In treated cells CB_2_ protein level was significantly decreased (*p* < 0.059) compared to controls (Figure 3a), whereas TRPV1 protein level was significantly increased (*p* < 0.05 vs. control), in keeping with qRT-PCR data (Figure 1a). Remarkably, protein expression of any other ECS element was unaffected (Figure 3b–g), while CB_1_ and DAGLα were below the detection limit.

To further investigate the possible effect of S1P on the endogenous levels of AEA and 2-AG, LC/MS analysis was performed. In keeping with protein expression of metabolic enzymes, no significant differences in the levels of these two eCBs were observed in cells treated with 1 µM S1P for 24 h (Figure 4).

### 2.2. Effect of Methanandamide on Mitochondrial Membrane potential

As shown by gene and protein expression data, C2C12 cells did express FAAH, the main hydrolytic enzyme of AEA (Figure 1, Figure 2 and Figure 3). Therefore, to rule out any interference of this enzyme activity, for subsequent cell treatments the non-hydrolysable, stable analogue of AEA methanandamide (mAEA) [40] was used. In particular, the mitochondrial membrane potential (ΔΨm) was evaluated in C2C12 cells treated with mAEA at different doses (2.5–5.0–10 μM) for 24 h, because of the key role of AEA on energy homeostasis [42,43]. ΔΨm values were measured with JC-1, a fluorescent dye that forms aggregates in the mitochondria with a red emission (~590 nm) but reverts to monomers with a green emission (~526 nm) when the mitochondrial membrane starts to depolarize [44]. Treatment for 24 h at lower doses of mAEA (2.5 μM and 5.0 μM) significantly (*p* < 0.01 and *p* < 0.05, respectively) increased ΔΨm values compared to controls (Figure 5). Yet, at the highest dose of 10 μM mAEA significantly (*p* < 0.05) decreased the ratio of aggregated to monomeric form of JC-1, by 48 ± 5.9%, compared to untreated controls (Figure 5).

The depolarization induced by 10 μM mAEA was further investigated in the presence of S1P, and interestingly mAEA failed to affect ΔΨm when cells were incubated in the presence of S1P, and vice versa S1P was unable to alter ΔΨm in the absence of mAEA (Figure 6). Taken together, these data suggested a possible crosstalk between the two lipid signaling systems. Moreover, to interrogate whether the mAEA depolarizing effect was mediated by eCB-binding receptors, the analysis was performed also in the presence of their selective antagonists: 0.1 μM SR1 for CB_1_ [45], 0.1 μM SR2 for CB_2_ [45], 0.1 μM ML193 for GPR55 [46], and 0.1 μM I-RTX for TRPV1 [47,48]. None of the antagonists affected ΔΨm when applied alone, yet pre-treatment with the specific antagonists of eCB-binding receptors prevented the depolarizing activity of 10 μM mAEA, whereas in the presence of I-RTX, selective TRPV1 antagonist, mAEA increased ΔΨm (#### *p* < 0.0001 vs mAEA alone) (Figure 6).

Since 10 μM mAEA markedly depolarized the mitochondrial membrane, we checked whether it could impair C2C12 cell viability. When added for 24 h, mAEA showed a dose-dependent cytotoxicity, and at 10 μM, it significantly reduced cell viability by 30 ± 3.0% (Figure 7).

In order to further analyze the effect of different concentrations of mAEA on mitochondrial activity, ΔΨm was measured in growing and serum-starved C2C12 myoblasts upon treatment for 24 h with mAEA at 5 µM, the dose immediately below the toxic one. For these assays, the Mitotracker Red CMX-Ros fluorescent dye was used, which stains mitochondria proportionally to their membrane potential. Laser-scanning confocal microscopy allowed the detection of fluorescence intensity, which was normalized to the cell number in each field. As shown in Figure 8, in 24 h serum starved C2C12 myoblasts ΔΨm significantly increased by 77.7 ± 7.5% compared to growing myoblasts (**** *p* < 0.0001), and it was further increased by 33.5 ± 2.9% (*** *p* < 0.001) upon challenge with 5 µM mAEA.

With the aim of interrogating whether S1P might affect mAEA activity on ΔΨm, the 5 µM mAEA challenge was performed also in the presence of 1 µM S1P for 24 h. Under these conditions, the mAEA-dependent increase in ΔΨm was fully reverted by S1P (Figure 9).

Moreover, based on the protein expression data, the role of eCB-binding receptors in 5 µM mAEA-induced increase in ΔΨm was studied. The specific blockade of TRPV1 by I-RTX abrogated the activity of 5 µM mAEA on ΔΨm, whereas SR1 and SR2 were ineffective (Figure 10).

### 2.3. Effect of Methanandamide on PGC1α, Respiratory Chain Complex Expression and Oxygen Consumption

With the goal to analyze the molecular mechanism involved in the effects of 5 µM mAEA on mitochondrial function, the expression of PGC1α and respiratory chain complexes was measured by Western blot analysis upon treatment for 24 h. As shown in Figure 11, the protein content of PGC1α was significantly enhanced, whereas that of the respiratory chain complexes were not affected by 5 µM mAEA challenge.

Moreover, the ability of 5 µM mAEA to affect oxygen consumption was examined by high-resolution respirometry. To this end, C2C12 myoblasts were treated with 5 µM mAEA for 24 h, then were collected and subjected to basal oxygen consumption (ROUTINE), proton leak (LEAK) and maximal oxygen consumption (E) measurement by the Oroboros-O2K system. At 5 µM, mAEA was unable to affect any of these respiratory parameters in C2C12 myoblasts (Figure 12).

## 3. Discussion

eCBs and S1P are bioactive lipids, which are critically important for the regulation of a plethora of key biological processes [8,13]. Intriguingly, these two lipid systems are composed by multiple enzymes that regulate their cellular tone, and specific receptor targets that transmit their effects, via a complex array of signal transduction pathways that remain only partly understood [8,13].

Here, unprecedented evidence is provided for a modulatory role of S1P on selected elements of ECS in cultured murine skeletal-muscle C2C12 cells, which are are widely used as a valuable model for interrogating, at the molecular level, key processes in skeletal muscle.

Importantly, S1P and eCB systems regulate fundamental biological events in skeletal muscle cell biology [10,35]. Indeed, S1P is known to activate muscle resident stem cells, favoring tissue repair by promoting also their proliferation and differentiation [49,50,51]. Moreover, it regulates tissue mechanical properties and responsiveness to insulin [10,52]. Myogenesis is also modulated by ECS, since CB_1_ expression has been reported to be up-regulated during myoblast differentiation and 2-AG was found to act via CB_1_ as endogenous repressor of myoblast differentiation, while mAEA treatment inhibited myogenesis [29,35]. The key role of the two families of lipid mediators in the control of skeletal-muscle cell biology is further underscored by the evidence that both signaling systems are dysregulated in a major skeletal-muscle disease, namely Duchenne muscular dystrophy [53,54]. In this context, CB_1_ was found to be overexpressed in murine dystrophic muscles, as well as in muscle biopsies of dystrophic patients [53], while muscles of dystrophic mice exhibited S1P lyase (SPL) upregulation and S1P deficiency, and accordingly pharmacological blockade of SPL in dystrophic mouse muscles exerted beneficial effects mediated by enhanced S1P, which in turn binds to and inhibits histone deacetylase [54].

In the present study, the possible functional interplay between the S1P and eCB systems has been explored in murine myoblasts by examining, as a first approach whether S1P treatment can modify receptor-mediated responsiveness to eCBs or the endogenous metabolism of these lipids. To this end, the expression of individual receptors and eCB-metabolic enzymes upon S1P challenge has been investigated both at mRNA and protein levels, and AEA and 2-AG cell content has been quantified. Notably, S1P at 24 h was found responsible of the selective up-regulation of TRPV1. Conversely, the protein content of CB_2_ was reduced by S1P, even though this variation was not mirrored by mRNA levels. In addition, mRNA levels of GPR55 receptor, as well as FAAH and DAGLβ enzymes in S1P-treated myoblasts were transiently increased by S1P at 24 h, and only GPR55 was found to be decreased at 48 h. It seems noteworthy that the present gene expression data on DAGL isozymes are consistent with the notion that non-neuronal cells (such as myoblasts) predominantly express DAGLβ, whereas DAGLα is more abundant within the brain [55]. In this context, it should also be noted that discrepancies between mRNA and protein expression of a specific target in treated cells are not quite unprecedented. Indeed, similar disparities between changes in mRNA abundance and protein content have been already reported [56], also within the ECS, by others [57] and by us [58,59]. Interestingly, an interaction between GPR55 and S1P5 receptor has already been demonstrated in a colon cancer cell line [60]. In particular, S1P5 strongly and specifically interacts with GPR55, and the activation of each receptor led to increased cell proliferation, ERK phosphorylation and cancer-associated gene expression. Conversely, co-activation of both receptors inhibited the above mentioned events, supporting the occurrence of a functional crosstalk [60]. In the herein-investigated model of murine myoblasts, S1P significantly rearranges cell sensitivity to eCBs without changing their endogenous content, suggesting a potential impact on the signal transduction driven by these lipid mediators.

Although important molecular effects of eCBs in skeletal muscle have been identified [53], so far it is not known if they engage the regulation of mitochondrial activity, which is essential for skeletal muscle metabolism and proper tissue plasticity. It should be recalled that AEA is known to play a key role in energy homeostasis [37,38,39], and indeed it can affect the shape and function of isolated mitochondria [42]. In line with this, some reports have even suggested a mitochondrial localization of CB_1_, supporting the view that AEA-dependent effects on energy production are triggered by these particular CB_1_ receptors [61,62]. Against this background, here we sought to ascertain whether the stable anandamide analogue mAEA could affect mitochondrial membrane potential, a key functional parameter of these organelles. Interestingly, our findings show that mAEA at the lowest concentration of 5 µM markedly enhanced ΔΨm, whereas the highest dose of 10 µM exerted an opposite effect. Moreover, in both cases the changes of ΔΨm elicited by mAEA were found to be mediated by TRPV1, since they were abrogated in the presence of its selective antagonist I-RTX. Likewise, treatment of mitochondria isolated from mice liver with higher concentrations (up to 50 μM) of AEA significantly decreased ΔΨm, down to ~70% of untreated controls, and they also increased membrane fluidity [42]. However, these effects were independent of CB_1_, CB_2_ or TRPV1 [42]. Similarly, in rat cortical neurons AEA, at the dose of 30 μM, was able to reduce mitochondrial membrane potential, induce the translocation of cytochrome c and activate caspase-3, overall leading to cell death independently of its receptor targets [63]. Instead, in human neuroblastoma SH-SY5Y cells, AEA dose-dependently, and again in the micromolar range, induced a mitochondria-dependent apoptosis, which in these cells was mediated by CB_1_ [38]. Overall, there seem to be different mechanisms by which AEA can affect mitochondrial function, possibly also in relation to cell type and species-specificity.

Moreover, the ability of S1P to counteract the depolarizing effect of mAEA in C2C12 cells appears to be noteworthy. Yet, S1P alone did not affect the mitochondrial membrane potential, which is in agreement with its ability to act as pro-survival factor and to control mitophagy and mitochondria dynamics [64,65,66]. Altogether, these results add a new dimension to the signaling pathways triggered by S1P and eCBs, and identify the Ca^2+^ channel TRPV1 as a pivotal target crucially involved in their crosstalk. The latter observation is quite remarkable, because so far most of the effects of eCBs in murine myoblasts have been ascribed to CB_1_/CB_2_ receptors with one exception describing TRPV1 as responsible for Ca^2+^ increase and subsequent promyogenic effect induced by cannabidiol [67]. The molecular mechanism by which S1P modulates responsiveness of myoblasts to eCBs has not been here investigated, paying special attention to the biochemical effects transmitted by S1P via eCB system, rather than its mechanistic action. C2C12 myoblasts are known to express four out of five S1P receptor subtypes, namely S1P1, S1P2, S1P3 and S1P4 [68], hence future studies will be necessarily addressed to dissect the exact role exerted by one or more receptor subtypes in the observed effects, even if it is conceivable to hypothesize that S1P2 is implicated. Indeed, this receptor subtype, although less represented than other isoforms, at least at mRNA level, [68], is regarded as the dominant receptor subtype in this cell type, being capable of transmitting key biological effects such as cell differentiation, inhibition of cell motility and cell proliferation [32,69].

Another relevant issue is the possibility that, instead of a cross-talk, mAEA and S1P may interact directly with TRPV1. Indeed, recent evidence supports the direct activation of TRPV1 by S1P in the context of pain and itch [70]. However, the hypothetic agonism of S1P at TRPV1 could not account for the observed up-regulation of TRPV1 at protein and mRNA level elicited by S1P at 24 h. Unfortunately, the analysis of the elevation of intracellular Ca^2+^ in transfected cells does not appear a suitable read-out to clarify this issue, because extracellular S1P itself, acting via its specific receptors, may potently induce such an elevation in many cell types, C2C12 myoblasts included [71]. More sophisticated approaches, such as silencing S1P receptor subtypes in C2C12 myoblasts, would be more informative, and should be the subject of independent investigations.

It is tempting to speculate that the herein-observed up-regulation of TRPV1 induced by S1P can counteract the increase in mitochondrial membrane potential induced by low mAEA, causing calcium overload that impairs mitochondrial function, thus mimicking the effect elicited by high dose of mAEA. In this context, it is well known that the large calcium gradient maintained across the mitochondrial inner membrane represents a critical signaling potential for this cation [72]. It is noteworthy that TRPV1 activation by capsaicin has been proven to increase free cytosolic calcium and improve energy metabolism by upregulating PGC1α in C2C12 myotubes, as well as in skeletal muscles [73]. Furthermore, it has been demonstrated that TRPV1 ligands decrease mitochondrial membrane potential and oxygen consumption in isolated mitochondria from rat cardiomyocytes [74]. Notably, in the present study, we found that low doses of mAEA are responsible for the increase in mitochondrial membrane potential and induction of PGC1α expression, without affecting oxygen consumption, whereas the highest dose of mAEA turns the TRPV1 action to negatively modulate mitochondrial membrane polarization and cell survival. The opposite functional changes mediated by TRPV1 may rely on concentration-dependent responses that via subtle increases in intracellular calcium positively affect mitochondrial membrane potential, whereas through mitochondrial calcium overload cause dysregulation of mitochondrial membrane potential and subsequent mitochondrial dysfunction. On a final note, markedly different consequences of low or high doses of AEA are not unprecedented. Indeed, in primary human melanocytes low (<1 μM) AEA leads to melanogenesis, whereas high (>5 μM) AEA causes development of melanoma tumors [75].

Overall, this study reveals a new critical cross-talk between S1P and eCB systems in skeletal-muscle cells, identifying the Ca^2+^ channel TRPV1 as a pivotal target and thus opening the avenue to new molecular approaches to control skeletal-muscle disorders characterized by calcium dyshomeostasis.

## 4. Materials and Methods

### 4.1. Materials and Reagents

Dulbecco’s modified Eagle’s medium, foetal calf serum and penicillin/streptomycin were from Corning (Corning, NY, USA). Bovine serum albumin (BSA) was purchased from Sigma-Aldrich (St. Louis, MO, USA). Sphingosine-1-phosphate (S1P, cod.62570) and methanandamide (N-(2-hydroxy-1R-methylethyl)-5Z,8Z,11Z,14Z-eicosatetraenamide, cod.157182-49-5) were from Cayman Chemicals (Ann Arbor, MI, USA); N-arachidonoyl-ethanolamine (anandamide (AEA), cod. A0580) and 2-arachidonoil-glycerol (2-AG, cod. A8973) were from Sigma-Aldrich (St. Louis, MO, USA). AEA-d8 and 2-AG-d8 were purchased from Cayman Chemicals (Ann Arbor, MI, USA) and were used as internal standards. The selective antagonists of CB_1_ SR141716A (SR1, cod. SML0800), of CB_2_ SR144528 (SR2, cod. SML1899) and of GPR55 ML193 trifluoroacetate (ML193, cod. SML1340) were from Sigma-Aldrich (St. Louis, MO, USA). The TRPV1 selective antagonist 5′-iodoresiniferatoxin (I-RTX, cod. 1362) was from TOCRIS (Bristol, UK). For molecular biology studies RevertAid H Minus First Strand cDNA Synthesis Kit, from Thermo Scientific (Waltham, MA, USA), and SensiFASTTM SYBR Lo-ROX kit, from Bioline (London, UK) were used.

### 4.2. Cell Culture and Treatment

Murine C2C12 myoblasts cell line (ATCC^®^ CRL-1772™) was grown in Dulbecco’s modified Eagle’s medium supplemented with 10% foetal bovine serum, 100 U/mL penicillin/streptomycin and 2 mM L-glutamine at 37 °C in a humidified 5% CO_2_ atmosphere [68]. When 90% confluent, myogenic differentiation was reached by substituting the proliferation medium with DMEM supplemented with 1 mg/mL BSA [68]. Then, cells were treated with 1 µM S1P [32] and/or with different concentrations (2.5, 5.0 and 10 μM) of mAEA for 24 or 48 h.

### 4.3. Quantitative Real Time-Reverse Transcriptase-Polymerase Chain Reaction (qRT-PCR) Analyses

For the quantification of gene-expression levels of receptors and metabolizing enzymes of eCBs, mRNA was extracted from control and S1P-treated C2C12 cells by using TRIzol, according to the manufacturer’s instructions (Life Technologies, Grand Island, NY, USA), and was quantified by using Thermo Scientific NanoDrop 2000 c UV-Vis spectrophotometer at 260 nm (Waltham, MA, USA). Subsequently, 1 μg of total mRNA was retrotranscribed in cDNA by using the RevertAid H Minus First Strand cDNA Synthesis Kit (Life Technologies, Grand Island, NY, USA). SensiFASTTM SYBR Lo-ROX kit was used to assess the relative abundance of CB_1_, CB_2_, GPR55, TRPV1, FAAH, DAGLα, DAGLβ, NAPE-PLD and MAGL on a 7500 Fast Real-time PCR System (Life Technologies, Grand Island, NY, USA), as described previously [31,76]. Primer sequences are reported in Table 1. The relative expression of different amplicons was calculated by the ΔΔCt method and converted to relative expression ratio 2^(−ΔΔ*Ct*)^ for statistical analysis [77]. All data were normalized to the endogenous reference genes GAPDH and β-Actin.

### 4.4. Western Blotting

Control and S1P- or mAEA-treated cells were lysed in RIPA buffer in the presence of a protease inhibitors cocktail (Sigma-Aldrich, St. Louis, MO). Then, they were sonicated three times and centrifuged at 9500 g for 15 min at 4 °C; supernatants were collected and protein content was determined by the Bio-Rad Protein assay (Bio-Rad Laboratories, Hemel Hempstead, UK). Cell lysates were mixed with Laemli sample buffer (heated for 10 min at 60 °C) so that an equal amount of protein per lane (20–70 μg) were subjected to a 10% sodium dodecyl sulfate–polyacrylamide gel electrophoresis (SDS-PAGE). Gels were then electroblotted onto polyvinylidene difluoride (PVDF) membrane (Amersham Hybond, GE Healthcare Life Science, Piscataway, NJ, USA). Subsequently, PVDF membranes were blocked with 5% milk, incubated overnight with the primary antibodies in cold room and then with the appropriate horseradish peroxidase-conjugated secondary antibody (cod.31461, Thermo Fisher Scientific, Waltham, MA, USA) for 1 h at room temperature. Primary antibodies specific for the following proteins were used: β-actin (cod.4970, Cell Signaling, Danvers, MA, USA), CB_1_ C-Terminal (cod. 10006590, Cayman Chemical, Ann Arbor, MI, USA), CB_2_ (cod.101550, Cayman Chemical), GPR55 (cod. 10224, Cayman Chemical), TRPV1 (cod. TA336871, OriGene, Rockville, MD, USA), FAAH (cod.101600, Cayman chemical), DAGLα (cod.PA5-23765, Invitrogen, Waltham, MA, USA), DAGLβ (cod.12574, Cell Signaling), MAGL (cod.10212, Cayman Chemical), NAPE-PLD (cod.10305, Cayman Chemical) and PGC1α (cod. SAB1411922, Sigma-Aldrich, St. Louis, MO, USA).

Total OXPHOS WB Antibody Cocktail (Abcam) was used to analyze complex II, III, IV and ATP synthase.

Detection was performed by Clarity Western ECL substrate (Bio-Rad, Hercules, CA, USA) as developed by Azure Biosystems c400 (Sierra Ct, Dublin, CA, USA). Immunoreactive band intensities was quantified by densitometric analysis through the ImageJ software (NIH, Bethesda, MD, USA), as reported [45].

### 4.5. LC/MS Analysis

The lipid fractions from control and S1P-treated C2C12 cells were extracted with chloroform/methanol (2:1 *v*/*v*) in the presence of internal standards (1 ng mL−1 AEA-d8, 200 ng mL−1 2-AG-d8). Then, a clean-up step was performed as previously reported [78]. Briefly, the organic phase was dried and then subjected to micro-solid phase extraction (µSPE) for a rapid clean-up, by using OMIX C18 tips from Agilent Technologies (Santa Clara, CA, USA). All analyses were performed on a Nexera LC 20 AD XR UHPLC system (Shimadzu Scientific Instruments, Columbia, MD, USA) with NUCLEODUR^®^ C18 Isis column from Macherey-Nagel GmbH and Co. (Neumann, Germany), coupled with a 4500 Qtrap mass spectrometer from Sciex (Concord, ON, Canada) equipped with a Turbo V electrospray ionization (ESI) source, operating in positive mode. The levels of AEA and 2-AG were then calculated on the basis of their area ratios with the internal deuterated standard signal areas, and their amounts in pmoles were normalized to the number of cells, as reported [75].

### 4.6. Mitochondrial Membrane Potential Assays

Mitochondrial membrane potential (ΔΨm) status of C2C12 cells, untreated or treated with mAEA at different concentrations (2.5 μM, 5.0 μM and 10 μM) for 24 h, was assessed with the lipophilic cation JC-1 [48]. Briefly, cells were plated onto a 96 well/plate at a density of 15 × 104. Specific and selective eCB-binding receptor antagonists SR1 (CB_1_), SR2 (CB_2_), ML193 (GPR55) and 5′-IRTX (TRPV1) were administered to cells at a concentration of 0.1 µM [46,47,48,79,80,81,82], 15 min before the incubation with mAEA, alone or in combination with 1 μM S1P for 24 h. As positive control, the potent mitochondrial uncoupler carbonyl cyanide m-chlorophenyl hydrazone (CCCP) was used at 50 μM. Then, cells were washed with PBS and stained with 2 μM JC-1 dye for 15 min, and then were analyzed by Enspire multimode plate reader (Perkin Elmer, MA, USA). Fluorescence intensities were calculated as red (590 nm) on green (526 nm) ratios, and were expressed as percentage (%) of controls as described [83].

The ΔΨm of C2C12 myoblasts was also assessed by Mito Tracker Red CMXRos (#M7512) probe (Thermo Fisher Scientific INC, MA, USA) and Laser scanning confocal microscopy after the treatment with mAEA (5 μM), in the presence of selective eCB-binding receptor antagonists, and/or 1 μM S1P for 24 h. Briefly, cells were seeded on microscope slides and grown for 24 h before being shifted in serum-free 0.1% BSA-containing medium in the presence or absence of mAEA. Specific and selective eCB-binding receptor antagonists were used as above described. The probe was diluted in DMEM medium without phenol red at a concentration of 50 nM, incubated for 30 min at 37 °C and 5% CO_2_ in the dark, and then slides were fixed in 2% paraformaldehyde. Subsequently, cells were washed twice and incubated with a permeabilization and quenching solution (0.1% Triton X-100 and ethanolamine (1:165) in PBS). A DAPI solution (#MBD0015, Sigma-Aldrich, MA, USA) was used to detect nuclei. Slides were mounted by using Fluoromount Aqueous Mounting Medium (Sigma-Aldrich, MA, USA), and images were obtained in a Leica SP8 laser scanning confocal microscope (Leica Microsystems GmbH, Wetzlar, Germany) using a 40× oil immersion objective.

### 4.7. MTT Assay

Cells were seeded into 96-well plates, at density of 1.5 × 104 per well, and incubated overnight. The day after, cells were incubated with different concentrations of mAEA (2.5–5.0–10 μM) for 24 h. Cell viability was assessed by the mitochondrial-dependent reduction of 3-[4,5-dimethylthiazol-2-yl]-2,5-diphenyl tetrazolium bromide (MTT; Sigma, St. Louis, MO, USA) to purple formazan. After 3 h, the MTT solution was discarded and 100 μL of DMSO were added to dissolve the formazan crystals. Cells were analyzed by Enspire multimode plate reader (Perkin Elmer, MA, USA). The cell viability was calculated by subtracting the 630 nm OD background from the 570 nm OD total signal of cell-free blank of each sample, and was expressed as percentage of controls set to 100%.

### 4.8. Intact Myoblast Respiration Using High-Resolution Respirometry

Oxygen consumption was analyzed in 2 mL glass chambers at 37 °C using the Oroboros oxygraph-2K high-resolution respirometer (Oroboros Instruments, Innsbruck, Austria) and substrate, uncoupler, inhibitor, titration (SUIT) protocols, as reported [84]. The oxygen flux normalized on the cell number is calculated as the negative time derivative of the oxygen concentration, measured in sealed chambers, and normalized on the instrumental background (measured in a dedicated experiment before assaying the cells). C2C12 myoblasts treated with 5 μM mAEA or vehicle (ethanol) for 24 h were subjected to respirometry analysis. After instrumental air calibration, 300.000 cells resuspended in DMEM with 1 mg/mL BSA were introduced into the chambers and the basal respiratory activity was measured as routine respiration (R). The LEAK state (L) represents the non-phosphorylating state of uncoupled respiration due to proton leak, proton and electron slip, and cation cycling [85] after the inhibition of ATP synthase by oligomycin administration (5 nM). The capability of the electron transfer system (ETS) was measured by uncoupler titrations using the uncoupler CCCP (1.5 µM/titration steps) as the readout of the maximal capacity of oxygen utilization (E). The residual oxygen consumption (ROX) that remains after the inhibition of ETS was determined by antimycin A (2.5 µM) injection. Data acquisition and analysis were performed using DatLab software (version 7.4, Oroboros Instrument, Innsbruck, Austria), and the oxygen fluxes recorded in the individual titration steps were corrected for ROX.

### 4.9. Statistical Analysis

Data were analysed by the GraphPad Prism 9.3.1 (471) program (GraphPad Software, La Jolla, CA, USA), and were reported as means ± S.E.M of 3 to 5 independent experiments. The statistical analysis was performed through the Student’s *t*-test and one-way or two-way analysis of variance (ANOVA), followed by Bonferroni post-hoc analysis. A level of *p* < 0.05 was considered statistically significant.

## Figures and Tables

**Figure 1 ijms-23-11103-f001:**
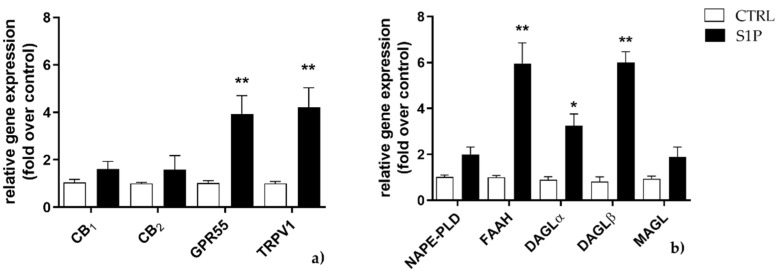
Effect of S1P on mRNA expression of endocannabinoid (eCB)-binding receptors (**a**) and metabolic enzymes (**b**) at 24 h. C2C12 cells were left untreated (in white) or were treated for 24 h with 1 µM of S1P (in black). The values are expressed as 2^(−∆∆^^𝐶𝑡)^ and normalized to β-Actin and GAPDH. Data are presented as means ± SEM (*n* = 5). Statistical analysis was performed by TWO-WAY ANOVA test followed by Bonferroni post hoc test. [* *p* < 0.05, ** *p* < 0.01 vs. control cells (CTRL)].

**Figure 2 ijms-23-11103-f002:**
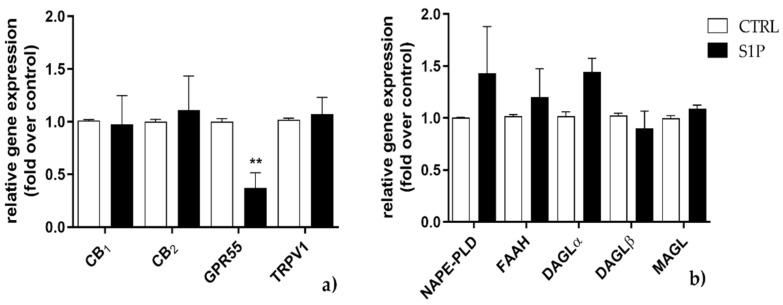
Effect of S1P on mRNA expression of endocannabinoid (eCB)-binding receptors (**a**) and metabolic enzymes (**b**) at 48 h. C2C12 cells were left untreated (in white) or were treated for 48 h with 1 µM of S1P (in black). The values are expressed as 2^(−∆∆^^𝐶𝑡)^ and normalized to β-Actin and GAPDH. Data are presented as means ± SEM (*n* = 5). Statistical analysis was performed by TWO-WAY ANOVA test followed by Bonferroni post hoc test. [** *p* < 0.01 vs. control cells (CTRL)].

**Figure 3 ijms-23-11103-f003:**
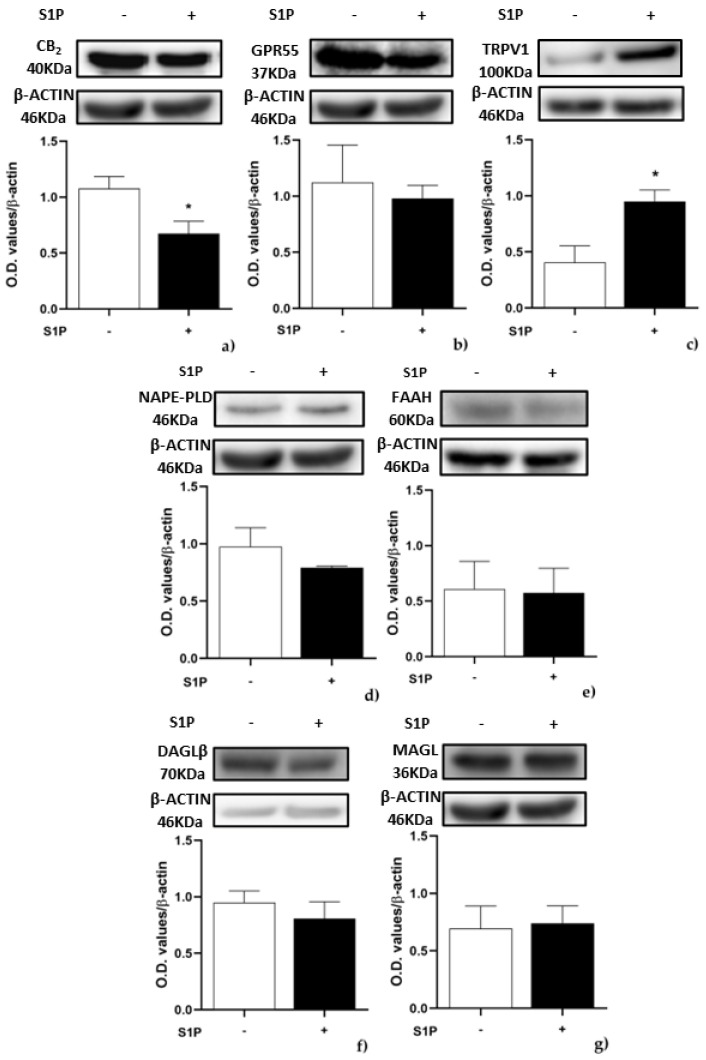
Effect of S1P on protein expression of endocannabinoid (eCB)-binding receptors and metabolic enzymes: C2C12 cells were left untreated (in white) or were treated for 24 h with 1 µM S1P (in black). Samples were subjected to Western blotting analysis using specific antibodies against the following proteins: (**a**) CB_2_; (**b**) GPR55; (**c**) TRPV1; (**d**) NAPE-PLD; (**e**) FAAH; (**f**) DAGLβ; (**g**) MAGL. Densitometric analysis values are expressed as relative optical density and normalized to β-Actin. The values represent the mean ± SEM of three independent experiments (*n* = 3). Statistical analysis was performed by ONE-WAY ANOVA test followed by Bonferroni post hoc test. [* *p* < 0.05 vs. control cells (CTRL)]. “+” with S1P, “−” without S1P.

**Figure 4 ijms-23-11103-f004:**
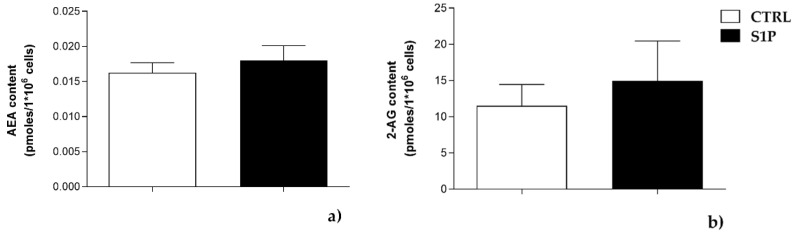
Effect of 1µM S1P on endogenous levels of AEA (**a**) and 2-AG (**b**), measured by using liquid chromatographic–mass spectrometry (LC-MS) analysis. The values represent the means ± SEM of three independent experiments (*n* = 3). Statistical analysis was performed by ONE-WAY ANOVA test, followed by Bonferroni post hoc test.

**Figure 5 ijms-23-11103-f005:**
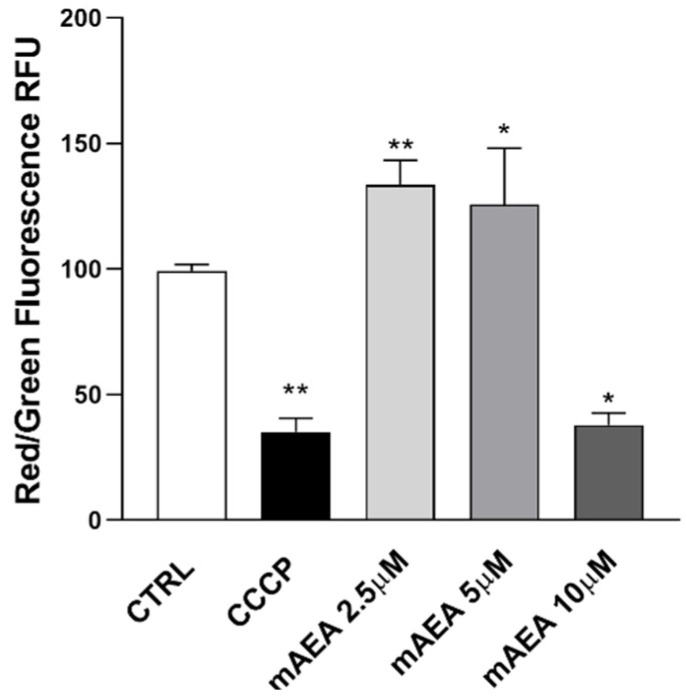
The polarization state of mitochondrial membrane was assessed after the cells were exposed to mAEA, at different doses (2.5, 5.0 and 10 μM) for 24 h, or to the oxidative phosphorylation inhibitor CCCP (50 μM) used as a positive control for (Δψm) decrease. Values are expressed as ratio of red (~590 nm)/ green (~529 nm) fluorescent intensity. Ratio are presented as mean ± SEM of three independent experiments (*n* = 3). [* *p* < 0.05, ** *p* < 0.01 vs. control cells (CTRL)].

**Figure 6 ijms-23-11103-f006:**
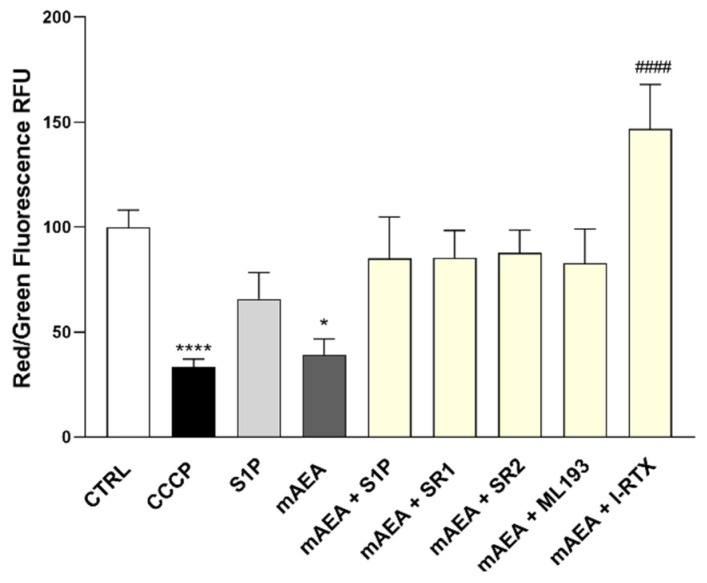
The polarization state of the mitochondrial membrane was assessed after the cells were exposed to mAEA (10 μM), and mAEA + S1P (1 μM) and in the presence of selective antagonists to eCB-binding receptors (SR1, SR2, ML193 and I-RTX) for 24 h. Values are expressed as ratio of red (~590 nm)/green (~529 nm) fluorescent intensity. Ratio are presented as mean ± SEM of three independent experiments (*n* = 3). Statistical analysis was performed by ONE-WAY ANOVA followed by Bonferroni post-hoc test [* *p* < 0.05, **** *p* < 0.0001 vs. control cells (CTRL); ^####^
*p* < 0.0001 vs. mAEA].

**Figure 7 ijms-23-11103-f007:**
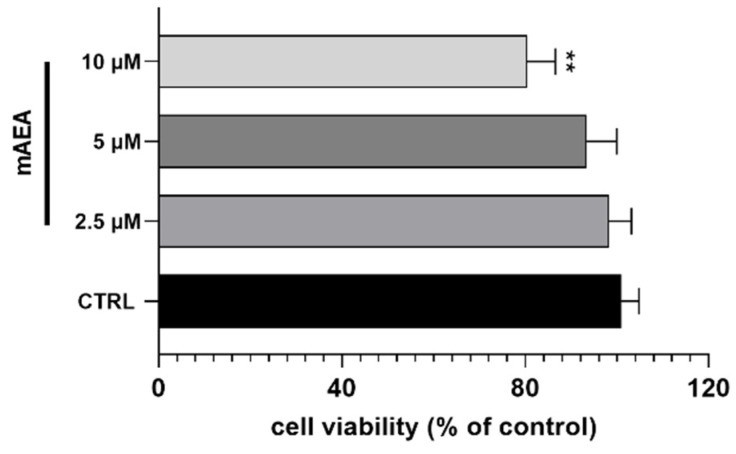
Assessment of cell viability using MTT assay on C2C12 myoblasts treated with mAEA at different doses (2.5, 5.0 and 10 μM) for 24 h. Values are expressed as % of control cells and presented as mean ± SEM of three independent experiment (*n* = 3). Statistical analysis was performed by ONE-WAY ANOVA followed by Bonferroni post-hoc test [** *p* < 0.01 vs. control cells (CTRL)].

**Figure 8 ijms-23-11103-f008:**
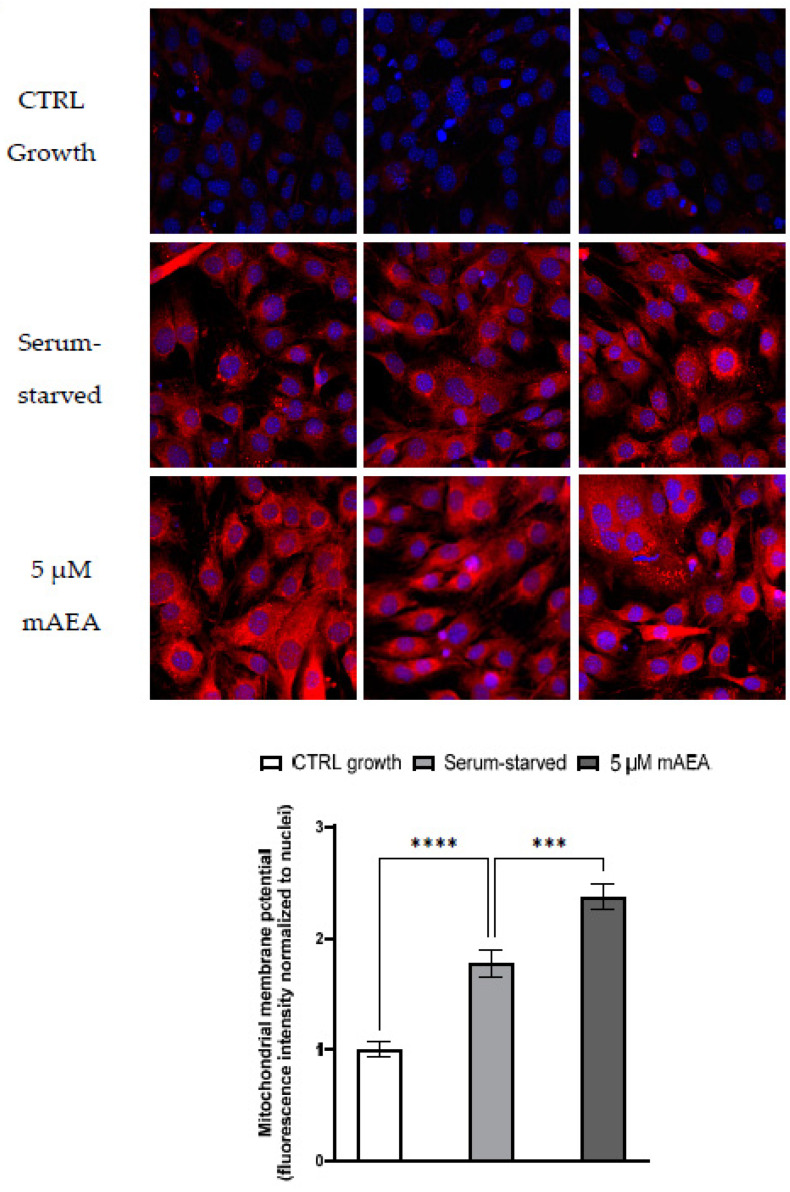
Effect of 5 μM mAEA on mitochondrial membrane potential. Confluent serum-starved C2C12 myoblasts were treated, or not, with 5 μM mAEA for 24 h and compared to growing (CTRL growth) myoblasts. Fluorescence intensity of Mitotracker Red CMX-Ros (Ex/Em: 579/599) was normalized to cell number and data are reported as mean ± SEM of six field per condition in each of the three independent experiments. Statistical analysis was performed by ONE-WAY ANOVA followed by Bonferroni post-hoc test [*** *p* < 0.001, **** *p* < 0.0001].

**Figure 9 ijms-23-11103-f009:**
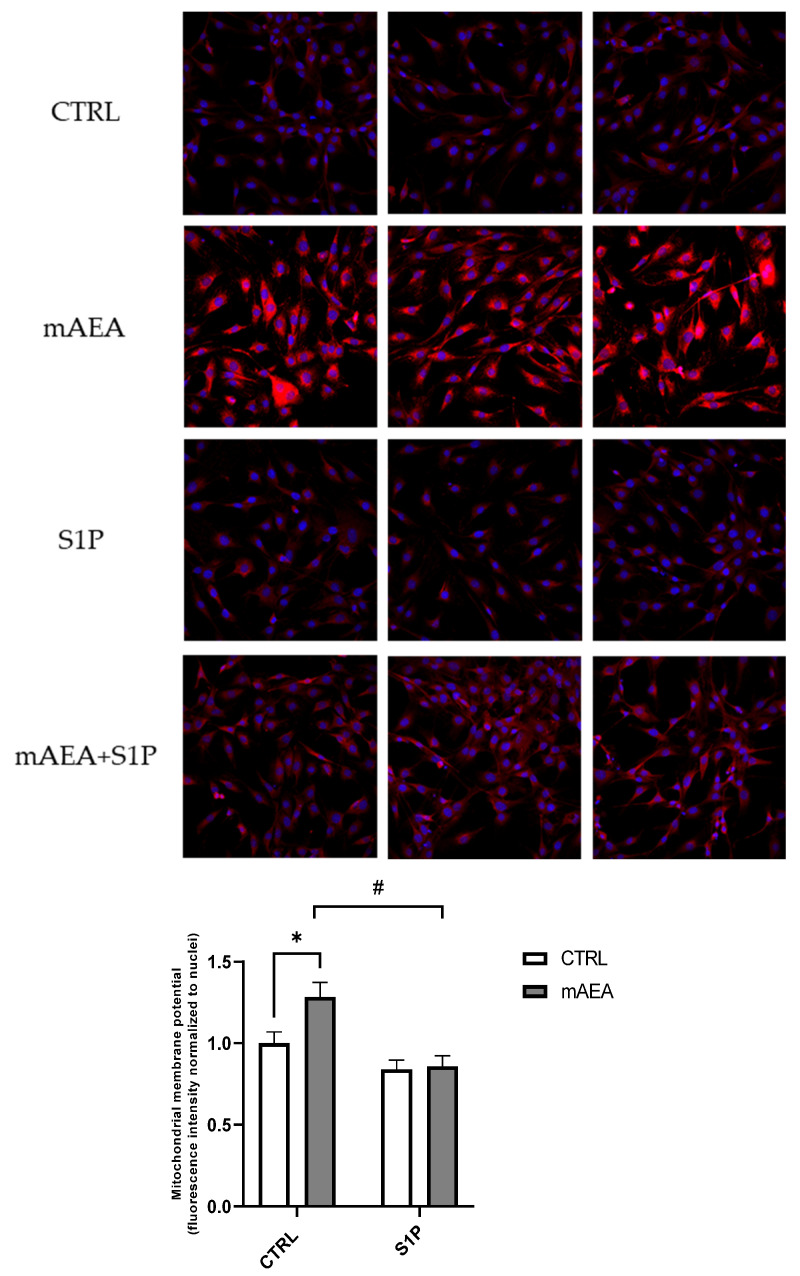
Effect of S1P on mAEA-induced mitochondrial membrane potential in C2C12 myoblasts. Confluent serum-starved cells were treated with 5 μM mAEA in the presence or absence of 1 µM S1P for 24 h. Fluorescence intensity of Mitotracker Red CMX-Ros (Ex/Em: 579/599) was normalized to cell number and data are reported as mean ± SEM of six field per condition in three independent experiments. Statistical analysis was performed by TWO-WAY ANOVA followed by Bonferroni post-hoc test [* *p* < 0.05 mAEA vs CTRL, # *p* < 0.05 S1P+mAEA vs mAEA].

**Figure 10 ijms-23-11103-f010:**
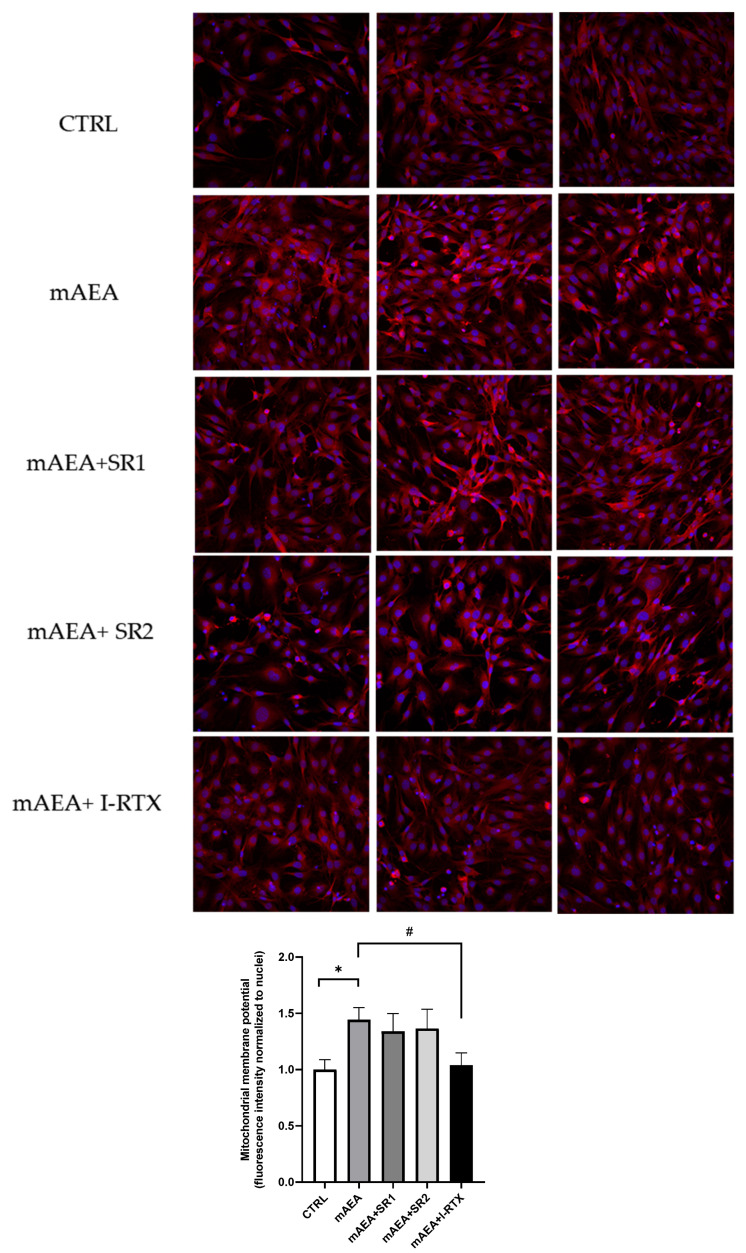
Role of eCB receptors on mitochondrial activity induced by 5 µM mAEA in C2C12 myoblasts. Confluent serum-starved C2C12 myoblasts were treated with 5 μM mAEA for 24 h in the presence of eCB receptor antagonists (SR1, SR2 or I-RTX). The fluorescence intensity of Mitotracker Red CMX-Ros (Ex/Em: 579/599) was normalized to cell number, and data are reported as mean ± SEM of six field per condition in each of the three independent experiments. Statistical analysis was performed by ONE-WAY ANOVA [* *p* < 0.05] and TWO-WAY ANOVA [# *p* < 0.05] followed by Bonferroni post-hoc test.

**Figure 11 ijms-23-11103-f011:**
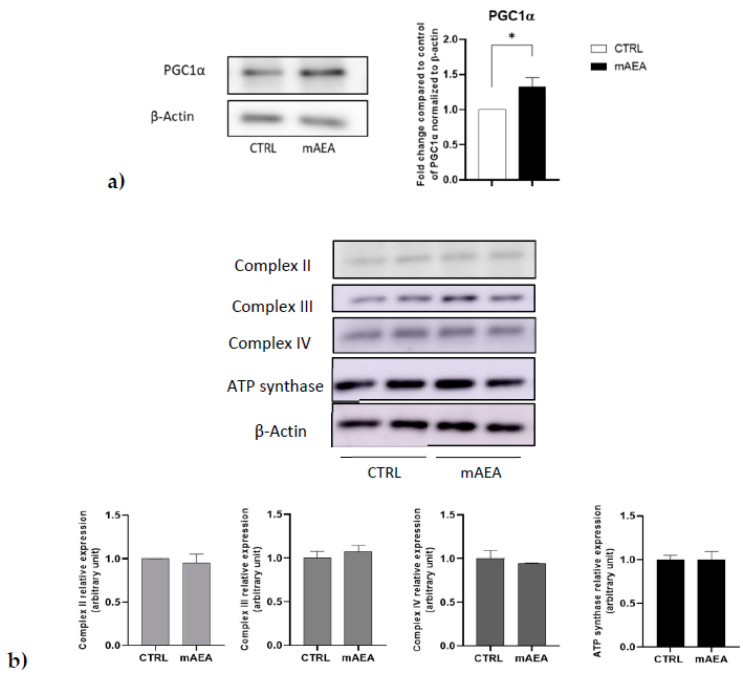
Effect of mAEA on PGC1α (**a**) and respiratory chain complexes’ (**b**) expression. C2C12 myoblasts were treated with mAEA (5 μM) for 24 h. Samples were subjected to SDS-PAGE and Western blot analysis using anti-PGC1α, as well as OXPHOS WB Antibody Cocktail specific for respiratory chain complexes and anti-β-actin antibody as housekeeping. The data were reported after densitometric analysis of the bands as means ± SEM normalized to β-actin of three independent experiments performed in duplicate. The data were reported after densitometric analysis of the bands as means ± SEM normalized to β-actin of three independent experiments. Statistical analysis was performed by analysis of variance using Student *t*-test [* *p* < 0.05 vs. control cells (CTRL)].

**Figure 12 ijms-23-11103-f012:**
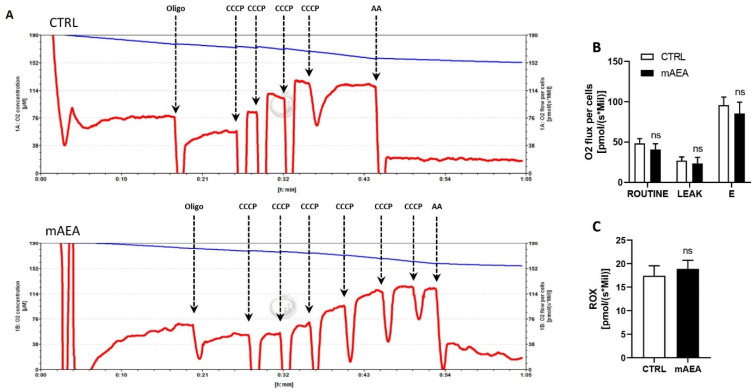
High-resolution respirometry of C2C12 myoblasts following mAEA treatment. C2C12 cells were treated or not with mAEA (5 μM) for 24 h. After detachment, myoblasts were subjected to high-resolution respirometry analysis by the Oroboros-O2K instrument. (**A**) Representative graphs of cell-respirometry analysis in the control (top) and treatment (bottom) conditions. The blue curve represents the oxygen concentration, whereas the red slope shows the oxygen consumption before and after the serial injections of oligomycin (Oligo), uncoupler CCCP and Antimycin (AA). (**B**) Bar chart graph of basal oxygen consumption (ROUTINE), proton leak (LEAK) and maximal oxygen consumption (E) values subtracted from residual oxygen consumption (ROX) in CTRL and mAEA treated cells. (**C**) ROX was also measured after AA administration. Data were reported as means ± SEM of the oxygen flux normalized on cell number of three independent experiments; ns= not significant.

**Table 1 ijms-23-11103-t001:** List of the primers used to assess the relative abundance of eCB-binding receptors and metabolic enzymes.

	Forward Primer (5′->3′)	Reverse Primer (5′->3′)
β-ACTIN	TGTTACCAACTGGGACGA	GTCTCAAACATGATCTGGGTC
GAPDH	AACGGGAAGCTCACTGGCAT	GCTTCACCACCTTCTTGATG
CB_1_	CCAAGAAAAGATGACGGCAG	AGGATGACACATAGCACCAG
CB_2_	TCGCTTACATCCTTCAGACAG	TCTTCCCTCCCAACTCCTTC
GPR55	ATTCGATTCCGTGGATAAGC	ATGCTGATGAAGTAGAGGC
TRPV1	TGAACTGGACTACCTGGAAC	TCCTTGAAGACCTCAGCATC
NAPE-PLD	AAGTGTGTCTTCTAGGTTCTCC	TTGTCAAGTTCCTCTTTGGAACC
FAAH	AGATTGAGATGTATCGCCAG	CTTCAGAATGTTGTCCCAC
DAGLα	AATGGCTATCATCTGGCTGAGC	TTCCGAGGGTGACATTCTTAGC
DAGLβ	TGTCAGCATGAGAGGAACCAT	CGCCAGGCGGATATAGAGC
MAGL	TTGTAGATACTGGAAGCCC	ATGGTGTCCACGTGTTGCAGC

## Data Availability

All experimental data are presented in the article.

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
