# Peer review of "The TRPV1 Receptor Is Up-Regulated by Sphingosine 1-Phosphate and Is Implicated in the Anandamide-Dependent Regulation of Mitochondrial Activity in C2C12 Myoblasts"

_ijms, 2022, doi:10.3390/ijms231911103_

Round 1
Reviewer 1 Report
The authors try to demonstrate a cross-talk between eCB system and SP1 in in murine C2C12 myoblasts.
Without ruling out the possible existence of a cross-talk between the two signalling systems, in my opinion, the direct action of SP1 on TRPV1 should be verified. It is possible that mAEA and SP1 interact directly with TRPV1, and this is not a cross-talk but an effect of the two ligands on the same receptors. For example, analysing the elevation of intracellular Ca2+ in transfected cells.
See: https://pubmed.ncbi.nlm.nih.gov/32089077/
Another consideration is that the cells were treated for 24 and 48 h with SP1. Does such a long time make metabolic sense? To check this, wouldn't it be better to test the effect of the treatment at 1 h, 2 h, 4 h, etc.?
Author Response
Response to Reviewer 1 Comments
Without ruling out the possible existence of a cross-talk between the two signalling systems, in my opinion, the direct action of SP1 on TRPV1 should be verified. It is possible that mAEA and SP1 interact directly with TRPV1, and this is not a cross-talk but an effect of the two ligands on the same receptors. For example, analysing the elevation of intracellular Ca2+ in transfected cells.
We thank the reviewer for his/her comment. Indeed, cross-talk between the two systems of lipid mediators can happen at different steps and the molecular mechanisms involved have not been investigated in detail in this study. However, the hypothesis formulated by the reviewer that S1P is able to drive the observed effects on eCB system via direct ligation to TRPV1, is indeed plausible in principle, but not in contrast with the proved occurrence of a cross-talk between the two signaling systems. It rather adds a further layer of complexity to the subject: the hypothetic agonism of S1P at TRPV1, for instance, could not account for the observed up-regulation of TRPV1 at protein and mRNA level elicited by S1P at 24 h. Since the involvement of TRPV1 in S1P action is a completely new finding of this study, the point raised by the reviewer is indeed interesting and worth of consideration. The suggested approach of analyzing the elevation of intracellular Ca2+ in transfected cells as read-out in our opinion appears to be not sufficiently informative, considering that extracellular S1P itself, acting via its specific receptors, is a potent inducer of intracellular Ca2+ increase in many cell types, C2C12 myoblasts included (see for instance Meacci et al. Sphingosine 1-phosphate evokes calcium signals in C2C12 myoblasts via EDG3 and EDG5 receptors. Biochem J 362:349– 357 (2002)).
Another consideration is that the cells were treated for 24 and 48 h with SP1. Does such a long time make metabolic sense? To check this, wouldn't it be better to test the effect of the treatment at 1 h, 2 h, 4 h, etc.?.
We thank the reviewer for his/her comment. Indeed, we believe that the study of the effect of S1P treatment at times shorter than the investigated ones is worth of interest. However, having in mind from the beginning to check at first whether S1P treatment could affect the expression at protein and/or mRNA level of the various members of eCB system (Figs 1-3), 24 and 48h incubations were adopted, presumably correctly. Since in these first experiments we obtained very exciting results, all the subsequent experiments have been performed employing long-term incubations with S1P. Taking into consideration that the subject of cross-talk between the two lipid messengers eCBs and S1P in skeletal muscle cells is quite broad, in this first manuscript we have tried to be especially focused, without varying the time of incubation with S1P throughout the study. Of course, we do not exclude that shorter incubations with S1P could affect eCB system and we are confident that this issue could be adequately addressed in future studies.
Reviewer 2 Report
In this study, the authors report on a cross-talk between S1P signaling and endocannabinoid signaling in C2C12 myoblast cells. They show that S1P stimulation of cells strongly increased the mRNA expression of the methanandamide receptors GPR55 and TRPV1, but not the classical CB1 and CB2. Also, the generating enzymes DAGLa+b, and the degrading enzyme FAAH are enhanced by S1P on mRNA but not on protein level. The S1P effect is transient and only detected after 24h, but not after 48h. They further show that methanandamide (mAEA) in high concentration is decreasing the mitochondrial membrane potential of cells and this effect is reversed by S1P cotreatment. They conclude that the TRPV1 is a key target in this cross-talk.
Major points:
1-It would be of great interest to better understand the mechanism by which S1P upregulated TRPV1 mRNA and protein expression, and reversed the high dose mAEA-triggered decrease of mitochondrial membrane potential. Does this indeed involve the S1P2 as hypothesized in the discussion part. Is it mimicked by a selective S1P2 agonist or can it be blocked by an S1P2 antagonist? It would be important to prove the receptor subtype involved and to prove that the effect of S1P is really mediated by receptor activation and not by receptor desensitization which may also occur by 1uM of S1P.
2-In Fig.11, the authors show that 5uM mAEA (they call this a low dose) increases PGC1a protein levels. How is this effect regulated by 1uM of S1P? In fact, the authors should be aware that mAEA has a maximal effect on TRPV1 already at 500nM, so the EC50 is even lower. Therefore, the here used concentrations of 2.5uM up to 10uM are all considered high.
3-In Figs 5 and 7, the y-axes are labelled as % of control, but the control columns are not at 100%.
Author Response
Response to Reviewer 2 Comments
1-It would be of great interest to better understand the mechanism by which S1P upregulated TRPV1 mRNA and protein expression, and reversed the high dose mAEA-triggered decrease of mitochondrial membrane potential. Does this indeed involve the S1P2 as hypothesized in the discussion part. Is it mimicked by a selective S1P2 agonist or can it be blocked by an S1P2 antagonist? It would be important to prove the receptor subtype involved and to prove that the effect of S1P is really mediated by receptor activation and not by receptor desensitization which may also occur by 1uM of S1P.
We thank the reviewer for his/her comment. We fully agree that understanding the molecular mechanism by which S1P drives the observed effects on eCB system is an interesting point. Indeed, we decided to hypothesize in the Discussion section the possible involvement of S1P2 receptor on this basis. Our experience in the characterization of S1P receptor subtypes involved in various biological/biochemical responses in different cell systems indicates that this task is rather complex. Even if, as pointed out in the Discussion section, S1P2 is a good candidate to transmit S1P action on eCB system, C2C12 cells express four out of the five S1P receptors and often, the molecular mechanism of S1P in this cell type implies a major role for one receptor subtype and a minor role for one or two other receptors (see for instance: Donati et al. FASEB J. 19,:449-51 (2005); Becciolini et al. Biochim Biophys Acta. 1761:43-51 (2006); Bernacchioni et al. Skelet Muscle. 2:15. (2012). This makes more complex the general approach to this issue, since it is inconclusive to proceed by examining exclusively the involvement of S1P2, and rather the study should include the four receptors expressed in murine myoblasts. Further complexity of this issue stems from the lack of selective agonists for each individual S1P receptor subtypes (for instance, S1P2 and S1P3), and from the unspecific effects exerted by the pharmacological antagonists available on the market (Salomone S, Waeber C. Front Pharmacol 2:9. (2011)). For all these reasons, in the numerous our previous studies aimed at identifying the involvement of S1P receptors in specific responses we have always performed additional and more informative experiments of specific gene silencing of the expressed receptors, that often have highlighted discrepancies in the results obtained by the two different experimental approaches (see for instance: Bigi et al. FEBS J. 2022 Jul 19. doi: 10.1111/febs.16579. Online ahead of print).
Since in this manuscript we are providing the first evidence for the occurrence of cross-talk between the two lipid mediator systems in cultured myoblasts, giving solid proofs of a modulatory role exerted by S1P onto ECB system, we are planning to develop this exciting and promising field by performing a systematic study of the molecular mechanisms involved in the action of S1P, in which the characterization of the role exerted by individual S1P receptors will be considered with primary attention, employing the multiple experimental approaches described above.
Thanks to this reviewer’s comment we have reconsidered the sentence in the discussion section dealing with the hypothetical involvement of S1P2 receptor.
New sentence in the discussion section:
“The molecular mechanism by which S1P modulates responsiveness of myoblasts to eCBs has not been here investigated, paying special attention to the biochemical effects transmitted by S1P via eCB system, rather than its mechanistic action. C2C12 myoblasts are known to express four out of five S1P receptor subtypes, namely S1P1, S1P2, S1P3 and S1P4 [69], hence future studies will be necessarily addressed to dissect the exact role exerted by one or more receptor subtypes in the observed effects, even it is conceivable to hypothesize that S1P2 is implicated. Indeed, this receptor subtype, although less represented than other isoforms, at least at mRNA level [69], is regarded as the dominant receptor subtype in this cell type, being capable of transmitting key biological effects like cell differentiation, inhibition of cell motility and cell proliferation [42,69]”.
2-In Fig.11, the authors show that 5uM mAEA (they call this a low dose) increases PGC1a protein levels. How is this effect regulated by 1uM of S1P? In fact, the authors should be aware that mAEA has a maximal effect on TRPV1 already at 500nM, so the EC50 is even lower. Therefore, the here used concentrations of 2.5uM up to 10uM are all considered high.
We thank the reviewer for his/her comment. Actually we deliberately omitted the study of the effect of S1P on the increase of PGC1a induced by mAEA. S1P has been reported to act as ligand of PPARgamma (Parham et al. FASEB J. 2015 Sep;29(9):3638-53) and thus capable of enhancing the expression of PGC1a. Moreover, acting via its receptors and PKA/CREB S1P has been shown to enhance the expression of the same protein. (Shen et al Cell Stress Chaperones. 2014 Jul;19(4):541-8.). We also observed that PGC1a expression is increased by S1P challenge of murine myoblasts (unpublished results). On this basis, the effect of S1P on mAEA-induced PGC1a expression could not be directly ascribed to the action of S1P on EC system.
Concerning the potency of anandamide on TRPV1, significant differences have been reported in literature, depending on cellular contexts (Ross RA, Anandamide and vanilloid TRPV1 receptors. Br J Pharmacol, 2003; 140(5): 790–801). For instance, in high expression recombinant cell lines using various methods, the potency (EC50) of anandamide has been measured in the range of 0.7 –5 μm. For instance, in native cell systems, the potency of anandamide ranges from 0.3 to 0.8 μM in blood vessels (relaxation) compared with 6 –10 μM in bronchus (contraction) and DRG neurons ([Ca2+]i and inward current) (Ross, 2003). In addition, in our previous work we found that mAEA at concentrations ≥5 μM can induce NHEM (primary human melanocytes) cell apoptosis through TRPV1 receptors (Pucci et al., Endocannabinoids stimulate human melanogenesis via type-1 cannabinoid receptor. JBC, 2012; 287(19):15466-78.). Of course, the definition of low and high doses in this study is referred to the investigated cell system and does not have general value.
3-In Figs 5 and 7, the y-axes are labelled as % of control, but the control columns are not at 100%.
We thank the reviewer for his/her comment. We have modified the figures in the revised manuscript: Figs 5 and 7 as mentioned by the reviewer, as well as Fig. 6 for the same reason.
Reviewer 3 Report
Review of the paper “The TRPV1 receptor is up-regulated by sphingosine 1-phosphate and is implicated in the anandamide-dependent regulation of mitochondrial activity in C2C12 myoblasts”, presented by S. Standoli, S. Pecchioli, D. Tortolani, etal.
The peer-reviewed article presents unprecedented evidence for a modulatory role of S1P on selected elements of endocannabinoid systems (ECS) in cultured murine skeletal muscle C2C12 cells that are widely used as a valuable model for interrogating at the molecular level key processes in skeletal muscle. The studies carried out by the authors are of considerable interest, since only a few studies have investigated Interactions between S1P and ECS in skeletal muscle, where both systems are active. Of particular importance are the results of experiments that studied the modulation of the mitochondrial membrane potential (ΔΨm) in the presence of S1P or antagonists to endocannabinoid-binding receptors. These experiments revealed a new critical cross-talk between S1P and eCB system in skeletal muscle cells, identifying the Ca2+ channel TRPV1 as a pivotal target and thus opening the avenue to new molecular approaches to control skeletal muscle disorderscharacterized by calcium dyshomeostasis.
This paper is well written, the introduction is clear, the aims and the hypothesis are correctly formulated, all the results obtained are expressively illustrated. The methods used by the authors are adequate to the task and are described in detail. The discussion is exhaustive.
The article should be published.
Author Response
Response to Reviewer 3 Comments
We really thank the reviewer, for his/her positive comments on the manuscript.
Round 2
Reviewer 1 Report
The new version of the manuscript only changes a small paragraph indicating that the role of different S1P receptor subtypes needs to be elucidated. As I indicated, the fundamental question is whether S1P acts through S1P receptors or interacts directly via TRPV1. There are results indicating that TRP channels are modulated by lipids, including S1P.
I believe that to publish these results, which state that SP1 regulates TRPV1 through a crosstalk between the S1P system and the eCB, is unwise. First it would be necessary to verify that by silencing the S1P receptors the results are different.
Although the authors give their opinion on the doubts I raised, nothing is reflected in the new version of the manuscript. I can understand that the authors do not want to consider new experiments, some of them long, but I do not find it appropriate that they do not discuss facts that are clear in the literature and that, depending on how, could invalidate the conclusions.
I think a more direct demonstration is essential to accept the publication of this paper.
Author Response
Reviewer 1 Comments
The new version of the manuscript only changes a small paragraph indicating that the role of different S1P receptor subtypes needs to be elucidated. As I indicated, the fundamental question is whether S1P acts through S1P receptors or interacts directly via TRPV1. There are results indicating that TRP channels are modulated by lipids, including S1P.
I believe that to publish these results, which state that SP1 regulates TRPV1 through a crosstalk between the S1P system and the eCB, is unwise. First it would be necessary to verify that by silencing the S1P receptors the results are different.
Although the authors give their opinion on the doubts I raised, nothing is reflected in the new version of the manuscript. I can understand that the authors do not want to consider new experiments, some of them long, but I do not find it appropriate that they do not discuss facts that are clear in the literature and that, depending on how, could invalidate the conclusions.
I think a more direct demonstration is essential to accept the publication of this paper.
Response to Reviewer 1 Comments
We do apologize for not having included her/his relevant point in our revised Discussion, since it merits to be part of a critical analysis of our data. Now this point has been included in the newly revised Discussion (page 14, lines 446-455), that now reads:
“Another relevant issue is the possibility that, instead of a cross-talk, mAEA and S1P may interact directly with TRPV1. Indeed, recent evidence supports the direct activation of TRPV1 by S1P in the context of pain and itch [71]. However, the hypothetic agonism of S1P at TRPV1 could not account for the observed up-regulation of TRPV1 at protein and mRNA level elicited by S1P at 24 h. Unfortunately, the analysis of the elevation of intra-cellular Ca2+ in transfected cells does not appear a suitable read-out to clarify this issue, because extracellular S1P itself, acting via its specific receptors, may potently induce such an elevation in many cell types, C2C12 myoblasts included [72]. More sophisticated approaches, like silencing S1P receptor subtypes in C2C12 myoblasts, would be more in-formative, and should be the subject of independent investigations.”
In addition, we would like to thank the reviewer for understanding how difficult and time-consuming it would be to perform new experiments, and hope that she/he can find it appropriate that we discuss in detail the issue raised.
Reviewer 2 Report
The authors have satisfactorily adressed all points of concern.
Author Response
We thank the reviewer for his/her positive comment.
Round 3
Reviewer 1 Report
The addition of a new paragraph opens up the possibility of a direct intraction, despite keeping the authors' hypothesis intact. I understand that fully clarifying what really happens would involve a very long work with no clear guarantees of success. In fact and having been proposed in other contexts, there is no certainty of direct intraction.
In the current version, the reader can reflect on this. I think the work deserves to be published.